# Simulation Study on Outdoor Wind Environment of Residential Complexes in Hot-Summer and Cold-Winter Climate Zones Based on Entropy-Based TOPSIS Method

**Xiang Liu, Wanjiang Wang \*, Zixuan Wang, Junkang Song and Ke Li**

School of Architecture and Engineering, Xinjiang University, Urumqi 830047, China;
107552201576@stu.xju.edu.cn (X.L.); 107552201582@stu.xju.edu.cn (Z.W.); junkangsong@stu.xju.edu.cn (J.S.);
107552101504@stu.xju.edu.cn (K.L.)
**\*** Correspondence: wangwanjiang@xju.edu.cn

**Abstract:** Driven by a large digital simulation environment, CFD calculation software is used to simulate test results so that they can be effectively applied to engineering practice. This paper explores the suitable outdoor wind environment for residential areas in the hot summer and the cold winter. Taking Xishan Huijing in Changsha as an example, the most unfavorable wind field environment is used as the boundary condition, and the optimal design mode for this residential area is explored based on the Butterfly platform. The research in this paper is mainly divided into five steps: (1) using Rhino 7.0 to establish a 3D model; (2) using the Butterfly 0.0.05 platform to simulate the wind field and export the data at the same time, and to realize the preview through the visualization method; (3) processing the exported simulation data and the calculation of related indices; (4) applying the entropy-based TOPSIS method on the MATLAB 2020 platform to rank the preferred scheme and obtain the corresponding index weights; and (5) using a K-means cluster analysis on SPSS 26 software to optimize the scheme. The results show that (1) the wind environment quality will be overestimated in the summer if the influence of neighboring buildings on the site is not considered, while the opposite is true in the winter, with the error of both reaching two times. (2) The weights of the indicators determined by the TOPSIS entropy weighting method indicate that wind protection in the winter should be prioritized over natural ventilation in the summer in this area. The maximum wind speed in the winter has the most significant weight, reaching 0.287, and the uniformity of the wind field in the summer is the most important, reaching 0.1102. (3) In the layout design of the residential district, the staggered layout of the 60 m high slab houses in the northern, northeastern, and northwestern directions of the site creates a better wind field environment, which attains the highest score by the TOPSIS entropy weighting method, reaching 0.1539, with the second highest score, reaching 0.1511, for the layout method. The research results will provide a scientific basis for the design of residential buildings in the hot-summer and cold-winter regions in China, and also help designers to better grasp the outdoor wind environment of residential buildings in the pre-design stage.

**Keywords:** hot-summer and cold-winter zone; residential community; parametric simulation; outdoor wind environment; entropy-based TOPSIS method



## 1. Introduction

With China's continuing urbanization [1], the increase in the number, density, and height of urban clusters has become a common trend in today's urban development [2]. In this case, the increase in the roughness of the urban subsurface causes the overall wind velocity in cities to decrease year by year, the frequency of static wind increases, the number of windy days to decrease, and the number of light windy days to increase [3]. The impact of wind on daily life in an area is mainly related to its comfort, safety, and cleanliness [4]. This study focuses mainly on pedestrian elevation, wind environment

optimization, and the assessment of the residential district layout. On the one hand, a good wind environment dramatically impacts the human living environment. A reasonable layout of residential areas is conducive to strengthening the natural ventilation, eliminating pollution, improving air quality, and ensuring the health and safety of residents. On the other hand, some studies have shown that natural ventilation and a comfortable outdoor thermal environment can significantly improve the livability and vitality of residential areas [5–7].

Some scholars propose that "Environmental Performance Design" should become an essential part of designing and optimizing urban spatial patterns [8]. Although the "Environmental Performance Design" trend has become inevitable, its implementation has many negative factors. Many existing scholars use numerical simulation tools (Ecotect, Phoenics, or ENVI-met) and optimization algorithms to optimize the analysis of the scheme to obtain early performance design points [9–14]. However, there are still severe challenges in applying the generation and optimization logic to outdoor wind field optimization. For one, the influence mechanism of outdoor wind fields on building shape is very complicated; it is physically dominated by the second-order nonlinear partial differential turbulence equation and is challenging to solve. For another, traditional housing planning and design is based on the climate zone and the wind rose diagram [15], and the design is usually based on the architect's personal experience, which is difficult to accurately quantify, and has a lot to do with the designer's experience and skills. In his previous research, Lin Borong also found that the methods, algorithms, and tools used to optimize the initial performance of structures still needed to be improved. He proposed a system solution of "model integration, performance evaluation, and interactive optimization" [16]. Therefore, it is foreseen that establishing a set of real-time interactive urban wind field evaluation and optimization methods will be the key to a city-air field study in the years to come.

In recent years, as the use of CFD has become mainstream, with more and more architects and planners are using digital simulation technology to optimize the design considerations of the settlement layout at the beginning of the design to limit the influence of the wind environment and increase the natural ventilation, in order to improve the comfort of human beings [4]. Based on this technology, scholars have conducted many relevant studies on the layout of residential buildings and the wind environment. For example, Shen and Ding [17] determined the optimum pattern for housing buildings by evaluating the effects on natural outdoor air ventilation. A housing planning model was developed in three Chinese municipalities, Beijing, Shanghai, and Guangzhou, and their correct orientation and spacing were determined. Similarly, To and Lam [18] analyzed the effect of various architectural categories and scales on the prevailing airfield and found that, when the building is perpendicular to the prevailing wind direction, a "canyon effect" occurs as the wind velocity increases. Huang et al. [19] discussed an ideal street model and analyzed the inclination angle of buildings and the influence of the inclination angle on the environment of the colonnade space. Wang used CFD methods to numerically simulate three representative hybrid housing blocks in Hefei, China, and proposed suitable floor heights and floor plans [20]. Huang et al. chose the wind velocity, wind pressure, wind velocity ratio, and other parameters by which to compare and analyze the typical layout of buildings in Hefei [21]. Hang et al. found that a point-to-point layout is the most effective way to ensure optimal wind conditions in dense building areas, followed by parallel, staggered, and closed layouts [22]. Hong et al. found that, when the wind direction is perpendicular to the linear building direction, most of the wind is blocked, and the low wind speed region occurs at the rear of the building [23].

In addition, climatic factors significantly impact the performance of buildings. However, due to the complexity of these climate factors, traditional building design is mainly based on expert opinion, and there needs to be a comprehensive analysis of the diversity of design solutions and their multidisciplinary performance. The current residential research for the hot-summer and cold-winter regions mainly includes the building façade [24], building envelopes [25,26], building airtightness [27], energy-efficient buildings [28–32],

and the distribution of outdoor tree species [33]. It can be seen that there is relatively little research on the residential layout in this climate zone, and the research results mainly focus on the energy consumption of individual buildings. Therefore, this article can serve as a methodological reference for the study of the outdoor climate environment of the residential layout in hot-summer and cold-winter areas.

Based on the above discussion, the objectives of this study include the following: (1) To assess the wind environment of residential neighborhoods in hot-summer and cold-winter regions in both winter and summer seasons using several metrics, including an assessment of wind field uniformity. (2) To establish a more reliable wind environment assessment model for residential communities using the entropy-based TOPSIS method. (3) To explore the optimal evaluation index for hot-summer and cold-winter areas. (4) Compare the advantages and disadvantages of the wind environment in different building layout situations. The research results can provide some reference for the spatial planning and construction of residential communities in hot-summer and cold-winter areas in the future.

The innovative points of this paper are:

(1) A research framework is provided, i.e., the analysis and application of multiple wind environment assessment indicators for the optimization of the height layout in different seasons and the processing and comparison of data.

(2) The selection of wind indicators for the outdoor wind environment study of settlements in hot-summer and cold-summer regions is prioritized; on the one hand, it can facilitate the selection of indicators for wind environment simulation by designers in the future, and on the other hand, it can help decision makers to make a choice when wind protection is required in winter and ventilation is required in summer collide.

(3) In the 3D modeling, the influence of the surrounding buildings on the model within the site is quantified in terms of the degree of the evaluation index, and the influence of the surrounding urban building background on the wind environment assessment is fully considered.

## 2. Research Methodology and Framework

### 2.1. Selection of Wind Environment Evaluation Indices

Due to the variability of the urban environment and the complexity of the outdoor wind environment, although scholars at home and abroad have performed a large quantity of study on the urban wind environment at pedestrian height, there are still differences among the indicators, and there is no unified standard yet. Due to the complexity of the outdoor environment, the use of a single wind speed index to describe the wind environment of the campus tends to be more one-sided, so this paper proposes to introduce five indices: wind speed, wind pressure, wind speed dispersion, comfortable wind, static wind, strong wind area ratio, and wind speed ratio to evaluate the wind environment research.

(1) Mean wind velocity: It is a commonly used evaluation metric in outdoor wind environments. It indicates the mean wind velocity at each point in the plane of the wind environment cross-section at a pedestrian height of 1.5 m, which can be used to measure the overall wind magnitude situation in the region.

(2) Wind pressure: China has explicitly regulated wind pressures on the windward and leeward sides of buildings, which are commonly used as assessment criteria in domestic outdoor wind environment studies, including "China Ecological Housing Technology" (GB/T 50378-2019), "Green Building Evaluation Standard" (GBT50378-2019), and "Green Design Code for Civil Buildings" (GB50180-93), etc., put forward the following relevant assessment indicators:

- In winter, in addition to the front row of buildings facing the wind, the wind pressure difference between the front and rear surfaces of a building should not exceed 5 Pa.

- In summer, more than 50% of external windows can be opened, and the wind pressure difference between the internal and external surfaces is more significant than 0.5 Pa.

This index reflects the ability of the outdoor wind environment to influence indoor ventilation.

(3)   Wind velocity ratio: This indicator better compares wind speeds in different areas and between building groups. The ratio of wind velocity in a specific area to its wind velocity at the same height not affected by the building, which reflects the degree of wind velocity influenced by the construction, independent of the initial wind velocity, and can better compare the wind velocity between different areas and different building groups, with the formula:

$$R_i = \frac{V_i}{V_0} \tag{1}$$

where $R_i$ is the mean wind velocity ratio, $V_i$ is the mean wind velocity in the area, and $V_0$ is the initial wind velocity without building disturbance.

Based on Tetsu et al. [34], the wind velocity ratio was used as an evaluation index for human comfort in a wind environment at pedestrian height, and the wind velocity ratio was defined as wind field comfort within the interval $0.5 \leq R \leq 2$.

(4)   Wind velocity dispersion: It is mainly used to evaluate the uniformity of the outdoor wind field. The concept reflects the variability of wind velocity in urban wind environments, which is used to characterize the dispersion of wind velocity in the observation area by standard deviation. A smaller value indicates a more uneven wind distribution in the field, and a more significant value indicates a more uneven wind distribution. The equation regarding wind velocity dispersion is as follows:

$$\sigma = \sqrt{\frac{1}{n}\sum_{i=1}^{n}(x_i - \mu)^2} \tag{2}$$

where $\sigma$ is the standard deviation of the data set, $x$ is the arithmetic mean of the data set; $i$ stands for sequence, $n$ is the sample size of the data set, and $\mu$ is the mean of all values in the data set.

(5)   Wind area ratio: some researchers, based on experimental results and calculations, have concluded that a speed of 1.04 m/s must be achieved at a walking height of 1.5 m to effectively increase the level of pollutants in the air, which means that in summer, speeds below 1.04 m/s are in the static wind zone and speeds below 0.5 m/s are in the wind shadow zone. In summer, a stronger wind velocity within the comfort range, but not exceeding the limit value of 5 m/s, is preferred. When the outdoor wind velocity is higher than 3–4 m/s, and the building faces the dominant wind direction, this is conducive to generating indoor penetrating wind. In winter, the wind velocity should be small, the weak wind zone should be significant, and the wind shadow zone should be small. According to the relevant standard, 2 m/s is the winter outdoor comfort zone's limit value. The range of wind speeds used in this paper and the definitions of static, comfortable, and strong winds are shown in Table 1.

To summarize, the indices selected in this paper involve a profound degree of evaluation, with commonly used indices of wind speed, normative indices of wind pressure, the range of wind speed reflecting human comfort or not, the uniformity of wind speed of the site from a macro perspective, and the degree to which the building is affected by wind speed so that it can be a better, more comprehensive, more systematic evaluation of the basic situation of the outdoor wind environment in the site.

**Table 1.** Strong wind, comfortable wind, static wind areas, and their definitions.

| | Wind Speed (Unit: m/s) | Define |
|---|---|---|
| Static wind | <1.04; | The ratio of the area of the region's static wind area to the total area measured; a larger ratio of the static wind area indicates that the region's air circulation and pollutant diffusion capacity are poor. |
| Comfortable wind | 1.04 < v < 5 (Summer); v < 2 (Winter); | The ratio of the area of the tested area to the total area where the wind speed meets the preset comfort criterion is the most intuitive evaluation index for studying the wind environment. |
| Strong winds | v > 5 (Summer); v > 2 (Winter); | A larger area ratio of the strong wind zone indicates that the wind speed in that zone is higher, making pedestrians feel uncomfortable or even causing wind damage. |

*2.2. Overview of TOPSIS-Entropy Method*

TOPSIS is often used to solve multi-objective decision-making problems [35]. Its basic idea is to use standardized matrices to find the optimal and worst solutions through the cosine method and then calculate the distance between the evaluated object and the optimal or worst solution to obtain the closeness between the evaluated object and the two. The operating principle of this method is relatively simple, and it can assess multiple research objects simultaneously. The evaluation results have high resolution and good practicality. Wang et al. [36] integrated the advantages of variable clustering and information theory for objective weighting of decision attributes to refine the accuracy of TOPSIS for building energy efficiency assessment. Seyyed Ali Sadat et al. [37] used the TOPSIS method to analyze the barriers to solar photovoltaic energy applications in Iran.

The entropy weight method means the information entropy of each indicator is calculated to examine the quality of the information provided by different indicators, which is used as a basis to calculate the weight of each indicator and provides an objective basis for comprehensive multi-factor evaluation. Qing Li et al. [38] used the entropy method to assess the energy efficiency of 18 groups of rural residential envelope retrofits, effectively evaluating the comprehensive performance of the retrofit solutions in terms of energy efficiency, carbon emissions, cost, and other multi-objective objectives. Yuan Qin et al. [39] used the entropy method to predict fly ash fiber concrete's durability and service life. Compared with subjective weighting methods, the entropy-based method avoids the bias of human subjective factors. The calculation formula is as follows:

$$r_{ij} = \frac{x_{ij}}{\sum_{i=1}^{m} x_{ij}} \tag{3}$$

$$e_j = -\frac{1}{\ln m} \sum_{i=1}^{m} r_{ij} \ln r_{ij} \tag{4}$$

$$d_j = 1 - e_j \tag{5}$$

$$\omega_j = \frac{d_j}{\sum_{j=1}^{n} d_j} \tag{6}$$

where $r_{ij}$ represents the ratio of $x_{ij}$ in $j$; $e_j$ represents the entropy value of each evaluation indicator; and $d_j$ is the value of information utility value.

Currently, scholars mainly use the entropy-based TOPSIS method to establish a hierarchy for assessing the level of construction development and a coupled and synergistic evaluation index system for the low-carbon construction industry and its supporting environment. Song et al. [40] screened the retrofitting solutions of rural buildings based on the TOPSIS method and obtained the preferred optimal solution and combined optimal solution containing each performance objective. Delgarm et al. [41] used multi-objective optimization techniques for architectural performance and indoor comfort were optimized, and the optimal solution was selected from the Pareto frontier using the TOPSIS method.

In a study [42], three MCDM methods (AHP), TOPSIS, and Choquet, were used to rank different façade design solutions. Chukwumaobi et al. [43] used the entropy-based TOPSIS method to select suitable PCM materials for timbre wall applications to improve performance.

In this study, the entropy-based TOPSIS method is combined to evaluate and analyze the wind environment indices in winter and summer seasons for different building heights in the summer hot and winter cold zone, then explore the best layout characteristics of settlements under this climate zone. The specific steps are as follows:

Step 1: Determine the required TOPSIS evaluation indicators. After completing the simulation of the wind environment, specific values of various evaluation indicators are obtained, and the TOPSIS indicators are determined according to actual needs.

Step 2: Determine the types of each indicator based on the specifications and the nature of the indicators. There are three types: extremely large, extremely small, and interval.

Step 3: According to the entropy weighting formula, obtain the weights of each indicator.

### 2.3. Research Framework

As shown in Figure 1.

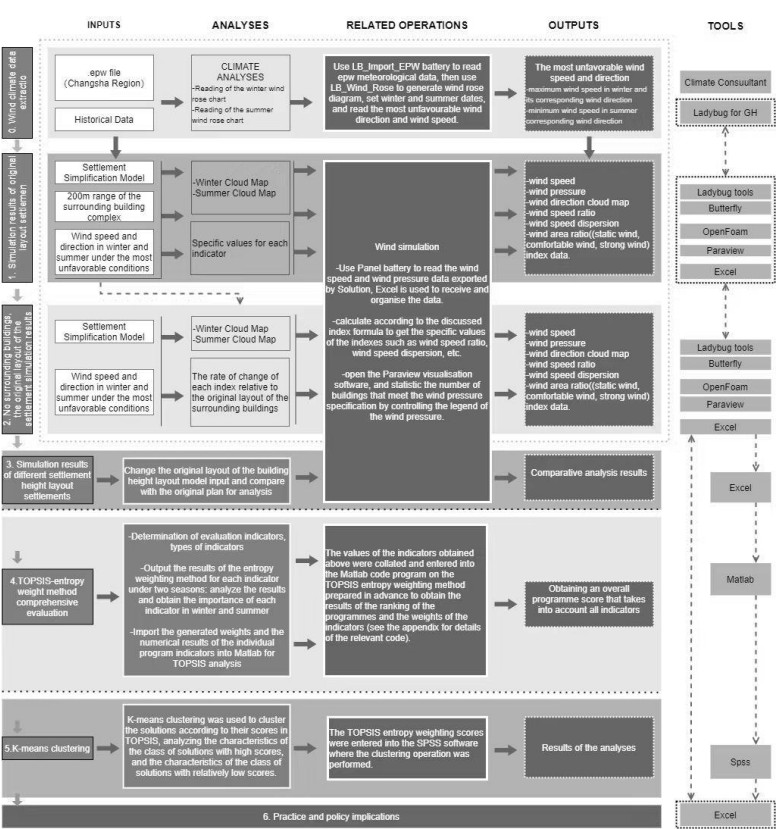

**Figure 1.** Research framework diagram.

## 3. Simulation Analysis

### 3.1. Introduction of Simulation Tools

The main workflow of this research is as follows: First, a 3D model of the simulated residential neighborhood is created using Rhino. Second, the geometry was imported into the Open Foam 5.0 software using Butterfly for CFD simulation and data exportation, and finally, the calculation results were visualized by Paraview 2017 software. This process can realize the simulation synchronized with the solution modification, and the optimal solution can be displayed quickly and accurately.

Compared with other leading CFD commercial simulation software such as Fluent and Phoenics, Open Foam is free and open-source. At the same time, its SnappyHexMesh function can perform hexahedral partitioning of the analysis mesh, which can better skin the analysis mesh and ensure the accuracy of the analysis. At the same time, Open Foam also supports multi-core parallel computing, which can shorten the simulation time. In 2012, Mostapha at the University of Pennsylvania developed a Ladybug plug-in on the Rhino Grasshopper platform, enabling users to call Open Foam directly from the Rhino. Open Foam is a visual programming method that further reduces the learning curve for users while allowing simultaneous parameter modification and simulation synchronization. On the other hand, Paraview is an open-source visualization software that enables visual representation of CFD results. It can handle billions of unstructured cells and more than trillions of structured cells, allowing a fast and clear presentation of simulation results through Paraview.

### 3.2. Selection of Simulation Sites and Simulated Climate Data

Changsha is located in the southeastern part of Asia and Europe, and its geographical location is from 111°53′ to 114°15′ east longitude and 27°51′ to 28°41′ north latitude. Since Changsha is located in the western part of the East Asian monsoon region, it is a humid subtropical climate with cold winters, hot and humid summers, and continental characteristics. The wind direction is characterized by an average wind velocity of 2.3 m/s and a northwest wind of 135°. Xishan Huijing, located in Yuelu District, Changsha, is a representative residential community with gardens, townhouses, and high rises. It covers an area of 80,607 m², with a green area ratio of 40%, a plot ratio of 2.3, and a building area of 192,836 m². The site mainly includes garden houses, townhouses, and high-rise apartment buildings.

The article selects the meteorological data from the China Building Thermal Environment Analysis Data Set. The data were compiled from meteorological data on the outdoor wind environment in Changsha from 1971 to 2003 and can better reflect the climate features of the city. By reading the local winter and summer wind rose diagrams (Figure 2), the most unfavorable wind velocity and the corresponding wind orientation are chosen for the simulation study: the maximum wind velocity in winter and the minimum wind velocity in summer. The meteorological information for each wind orientation is listed in Table 2. The most unfavorable wind velocity in summer is 2.36 m/s, corresponding to the south wind direction; in winter, the most unfavorable wind orientation is southeast, with a wind velocity of 2.05 m/s.

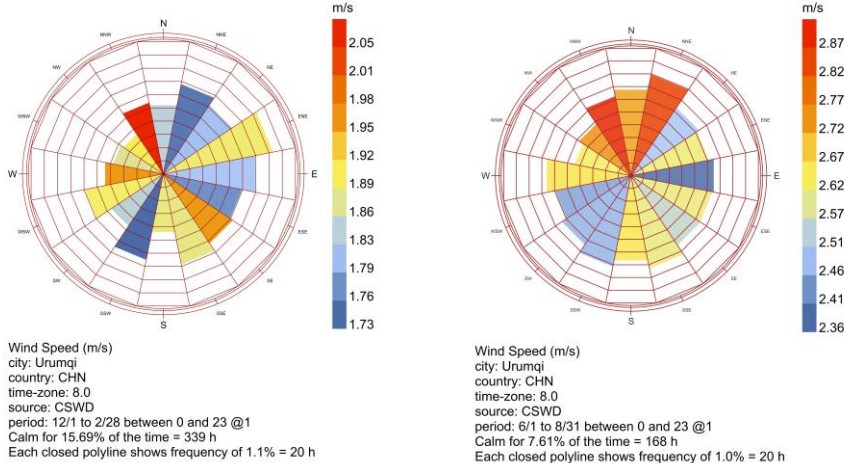

**Figure 2.** Winter and summer seasonal roses.

**Table 2.** Selected wind velocity and the corresponding wind orientation in winter and summer.

|         | Most Unfavorable Wind Speed | Corresponding Wind Direction |
| ------- | --------------------------- | ---------------------------- |
| Winter  | 2.05 m/s                    | 337.5°                       |
| Summer  | 2.36 m/s                    | 90°                          |

### 3.3. Simulation Model Construction and Parameter Settings

When building numerical models, the influence of model boundary effects is ignored, which can easily lead to excessive wind speeds and inaccurate thermal environments. To mitigate this exaggeration, many studies have used the surrounding environment model as a buffer zone around the study area. In this thesis, Rhino 7.0 modeling software is used to simplify the drawing of 18 single buildings and all surrounding buildings within 200 m of the perimeter (e.g., Figure 3), with dimensions of 479 m × 236 m × 60 m. According to the relevant research experience, the distance from the building cluster's top to the computational domain's boundary should be greater than 5H. The horizontal distance from the cluster's outer edges to the boundary of the computational domain should be greater than 5H so that the determined final size of the computational domain is 1662 m × 1007 m × 295 m. The simulation model is constructed by setting the simulation parameters through the Butterfly plug-in of the Grasshopper platform and calling the wind environment simulation engine OpenFOAM to carry out the numerical simulation, which lays the foundation of the validation analysis.

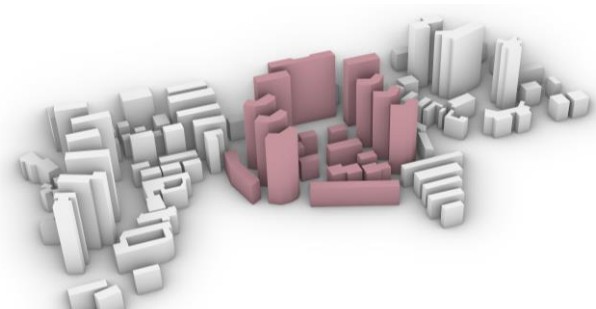

**Figure 3.** Construction of 3D Geometric modeling of the target and surrounding buildings.

Based on the completion of the geometric model construction, a numerical simulation model of the wind environment is constructed through the Grasshopper plug-in Butterfly, and the main simulation parameters, such as mesh division and boundary condition settings, are set.

(1) Mesh division

The uniform ortho-hexahedral and polyhedral structured meshes are generated by the BlockMesh module, and the values of the $x$-axis, $y$-axis, and $z$-axis are set to be 80, 80, and 40, respectively, considering the simulation time required. SnappyHexMesh is then used to automatically perform surface fitting refinement of the mesh at the height of 1.5 m of the pedestrian height at the wall surface of the building. (When generating a mesh, the relevant parameters in the castellated-MeshControls, snapControls, addLayersControls, and meshQualityControls subdictionaries of SnappyHexMesh are used to control mesh eigenface cuts, re-refinement, alignment, boundary layer meshQualityControls. See the Supplementary Materials Table S1 for more information and parameter settings). To adapt to changes in the flow field, while the outer region is arranged with a coarser grid, forming a dense inner and sparse outer grid structure, as shown in Figure 4.

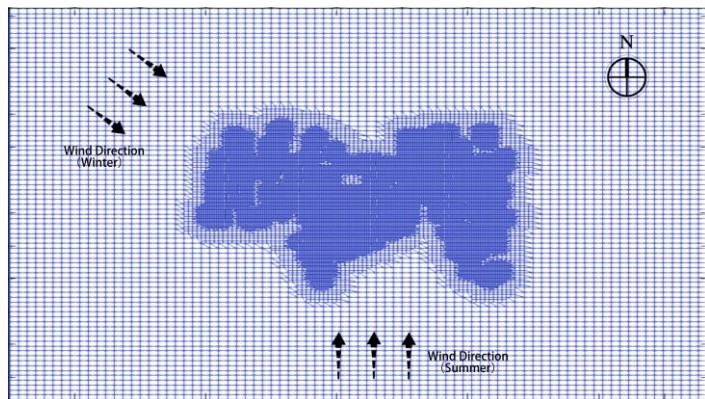

**Figure 4.** Mesh division.

(2)    Boundary conditions

In Butterfly, the boundary conditions required for the simulation of the outdoor wind environment are relatively simple, generally including the boundary of the building wall and the boundary conditions around the wind tunnel set according to the computational domain. Therefore, in this paper, the chosen ground roughness is the general roughness 0.25. In addition to this, at the entrance of the computational domain, the most unfavorable wind speed and direction data are used as the inflow boundary conditions. At the exit, the outflow boundary condition with zero gradient is specified and the top and side surfaces are set as free slip conditions. Different wall functions are used for different wall types. The specific editing condition parameter settings are shown in Table 3.

**Table 3.** Boundary conditions.

|  | General Purpose Wall | Ground Level | Side/Top | Exits | Entrances |
|---|---|---|---|---|---|
| V (Velocity) | fixedValue | fixedValue | slip | inletOutlet | atmBoundary LayerInletVelocity |
| P (Pressure) | zeroGradient | zeroGradient | slip | fixedValue | zeroGradient |

(3)    Other parameters

Simulations were performed using OpenFOAM 4.0 simulation software, with parameters called and adjusted through the Butterfly.0.0.64 interface of the Grasshopper parameterization platform. The 3D steady-state RANS equations and a realizable k-ε turbulence model, which performs well in urban wind flow simulations, were used for all simulations.

## 4. Results

### 4.1. Layout Representation

To study the influence of building height in a layout on the actual wind environment of the community and then determine the most suitable building height layout, the most unfavorable wind velocity and wind orientation in winter and summer were selected for analysis. By changing the building height of the region and controlling other relevant parameters, average wind velocity, wind pressure, wind velocity dispersion, comfortable wind, static wind, strong wind area ratio, and wind velocity ratio were observed to compare and analyze the most suitable height layout form of the area. The height of the building is taken from the number of residential floors commonly found in the area, namely 6 (18 m), 11 (33 m), and 18 + ground floor (60 m). Combining English and Arabic numerals refers to a particular building layout form to facilitate observation and analysis. As is shown in Figure 5 and Table 4, there are nine different height layout forms of residential areas.

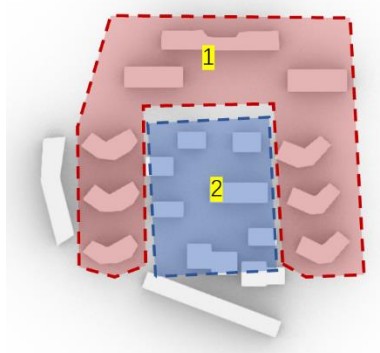

**Figure 5.** Areas represented.

**Table 4.** Zoning and simulation values of Changsha Xishan Huijing neighborhood.

|  | A1 | B1 | C1 |
|---|---|---|---|
| A2 | A1A2 (1, 2 area building heights are 18 m) | A2B1 (33 m for area 1 and 18 m for area 2) | A2C1 (60 m for area 1 and 18 m for area 2) |
| B2 | B2A1 | B2B1 | B2C1 |
| C2 | CA1 | C2B1 | C2C1 |
| A: Building height 18 m<br>B: Building height 33 m<br>C: Building height 60 m | | | |

### *4.2. Data Analysis*

To facilitate the control analysis, the existing scheme, i.e., the outer circle is 60 m, and the inner circle is 18 m, is set as the blank control group. The remaining eight schemes are compared with this scheme one by one for the wind environment evaluation index.

#### 4.2.1. Blank Control Group

We perform numerical simulation of the original scheme, and the simulation cloud results are shown in Table 5, and the related data results are shown in Table 6.

**Table 5.** Option 1: A2C1 settlement layout in 1.5 m out of the wind environment cloud map (W represents winter and S represents summer).

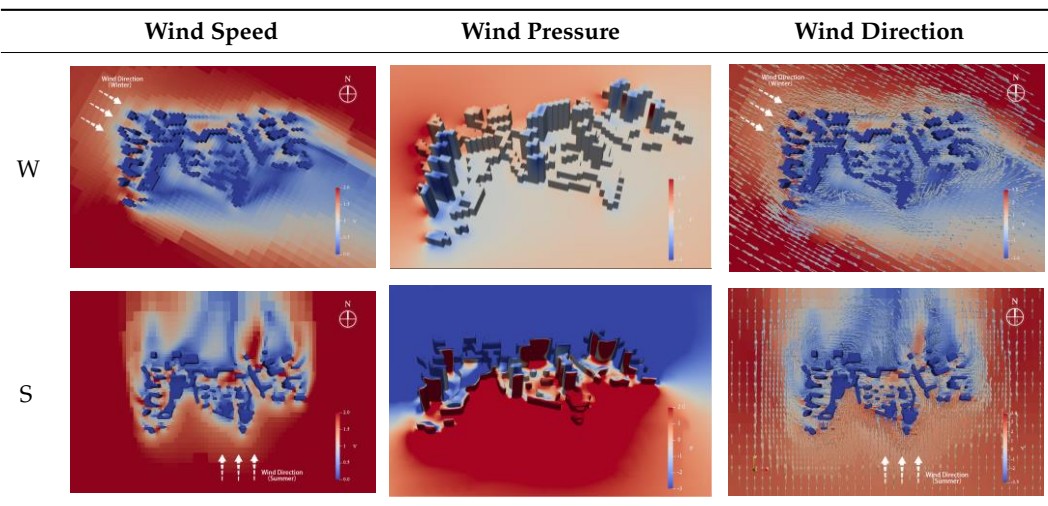

| | Wind Speed | Wind Pressure | Wind Direction |
|---|---|---|---|
| W | | | |
| S | | | |

**Table 6.** Wind environment evaluation index data related to the layout of A2C1 settlements.

| Seasons\Index | $V_{max}$ | $\overline{V}$ | $R_{max}$ | $\overline{R}$ | $\overline{P}$ |
|---|---|---|---|---|---|
| Winter | 2.175 | 1.314 | 1.061 | 0.641 | 0.266 |
| Summer | 2.654 | 1.605 | 1.124 | 0.680 | 0.270 |
| Seasons\Index | $\sigma$ | $A_{static}$ | $A_{comfort}$ | $A_{strong}$ | $P_{max}$ |
| Winter | 0.634 | 0.345 | 0.938 | 0.062 | 3.054 |
| Summer | 0.696 | 0.243 | 0.757 | - | 3.866 |

The simulation results show that the maximum wind velocity in the settlement layout form of A2C1 is 2.175 m/s in winter and 2.654 m/s in summer, and the cloud diagram indicates that the maximum wind velocity in winter mainly appears at the rear of the slab-sided high rise; in summer, the maximum wind velocity appears outside in the high-rise slab-type buildings. In terms of the wind field area, in this layout, the maximum wind velocity does not exceed 5 m/s in summer; the site with wind velocity greater than 2 m/s in winter accounts for 6.2%; among them, the static winter wind area accounts for 34.5%, and the summer static wind area accounts for 24.3%; the static winter wind area is mainly concentrated in the middle of the multi-story building group, and the summer static wind zone is mainly in its rear part; and the corresponding comfortable wind field, that is, the area less than 2 m/s in winter accounts for 93.8%. It means that the comfortable wind field accounts for a relatively large proportion; the most suitable wind field in summer is 1.04 m/s < v < 5 m/s, accounting for 75.7%.

From the maximum wind velocity ratio, we can find that in this layout form, the wind velocity in winter and summer is within the comfort range of [0.5, 2] due to the influence of building changes. In addition, from the mean wind velocity ratio and wind velocity dispersion, the wind field is more uniform and comfortable in winter than in summer under this layout form. From the average wind velocity ratio, this layout has little effect on the airflow in winter and summer, probably because the distance between the buildings is reasonable in this layout, which will not produce large wind velocity owing to the "funnel effect" and "canyon effect".

Through the statistical analysis of the final simulation results, it is found that under this arrangement, the wind pressure of the buildings in the site before and after construction in winter is lower than 5 Pa, which meets the requirements of the national building design code. There are 10 buildings with the wind pressure difference between the front and rear of the buildings above 0.5 Pa in summer; according to the national standard proposed that more than 75% of the buildings in the site need to meet this standard, which means that at least nine buildings need to meet the requirement that the pressure difference between the front and rear of the building is more significant than 0.5 Pa. This scheme meets the national requirements for the wind pressure difference between the front and rear of the building in winter and summer. In addition, as shown in the cloud diagram, the wind pressure distribution in winter is more uniform overall than in summer. As reflected by the wind direction cloud diagram, under this layout, there is no vortex in winter; the summer wind also shows a certain orderliness and is less susceptible to a vortex.

Next, we remove the surrounding buildings to perform another simulation, which is shown in Table 7, and the numerical results are shown in Table 8.

**Table 7.** Wind environment clouds of the control scenario without the influence of surrounding buildings (W represents winter and S represents summer).

| | **Wind Speed** | **Wind Pressure** | **Wind Direction** |
|---|---|---|---|
| W | | | |
| S | | | |

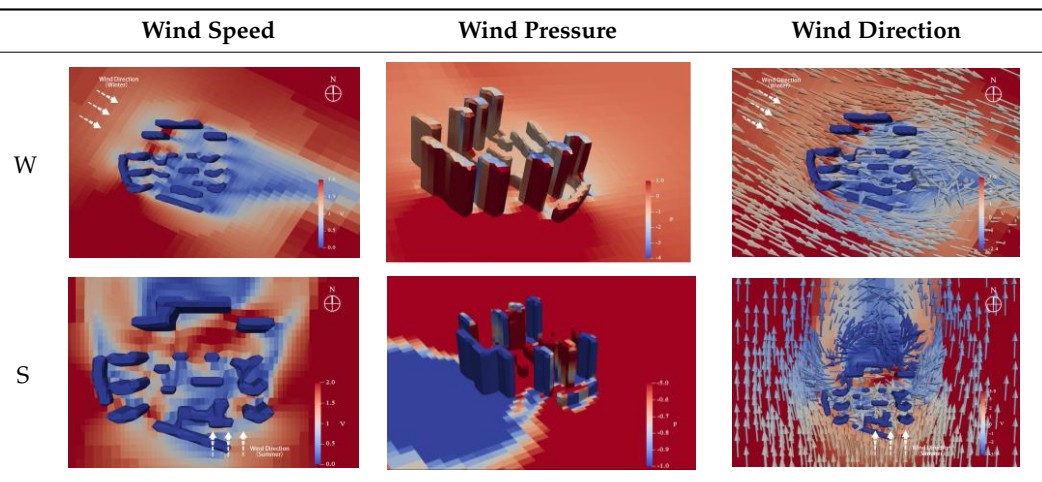

**Table 8.** Rate of change in relevant wind environment evaluation indicators for the control scheme without the influence of surrounding buildings.

| Seasons\Index | $V_{max}$ | $\overline{V}$ | $R_{max}$ | $\overline{R}$ | $\overline{P}$ |
|---|---|---|---|---|---|
| Winter | 0.057 | 0.364 | 0.057 | 0.348 | 0.086 |
| Summer | $-0.049$ | 0.279 | $-0.048$ | 0.222 | $-0.207$ |

| Seasons\Index | $\sigma$ | $A_{static}$ | $A_{comfort}$ | $A_{strong}$ | $P_{max}$ |
|---|---|---|---|---|---|
| Winter | $-0.379$ | $-0.792$ | $-0.214$ | $-0.764$ | $-0.179$ |
| Summer | $-0.191$ | $-0.626$ | 0.201 | - | $-0.058$ |

To facilitate comparative analysis with the rising and falling trends of the blank control group, the values of the magnitude of the statistical changes were used; the equation for the solution was

$$\Delta = \frac{N'}{N_0} - 1 \tag{7}$$

From the results of the simulation, we can see that the model of the surrounding buildings was not included in the data simulation, which has a much more significant effect on the wind environment in winter than in summer, mainly in terms of wind protection in winter. In winter, the wind velocity of the site increases, the maximum wind velocity increases, the static wind area decreases, the vital wind area increases, and the comfortable wind area decreases. At the same time, the degree of influence of the buildings on the wind velocity and the most unfavorable degree tend to increase. From the wind pressure index, the average wind pressure increases more in winter, and the maximum wind pressure decreases. In winter, according to the requirement of the Green Building Evaluation Manual that the wind pressure difference between the front and rear of the building is no more than 5 Pa, five buildings exceed the requirement of the code. The generated vortex shows that the intensity is larger and linearly distributed near the air outlet.

The simulation results will be overestimated in summer if the surrounding environment is not considered. As far as the wind velocity index is concerned, the maximum wind velocity decreases less, and the average wind velocity increases more; the static wind area is reduced significantly, which increases the comfort of indoor air. Meanwhile, the uniformity of wind velocity throughout the site is more reasonably improved. As for the wind pressure index, the average wind pressure and maximum wind pressure have a decreasing trend. As far as the buildings are concerned, the number of buildings meeting the standard has risen to 14, which is a significant improvement over the original scheme. The vortex cloud diagram shows that in the absence of building interference, the vortex clouds in summer

mainly appear behind the high-rise floor slabs, and the wind field inside the site is more complex.

### 4.2.2. Comparative Analysis of Scenarios

Option 2: When the height of the buildings in the site is 18 meters, the results of the simulated cloud maps are shown in Table 9 and the numerical results are shown in Table 10.

**Table 9.** Option 2: A1A2 settlement layout in 1.5 m out of the wind environment cloud map (W represents winter and S represents summer).

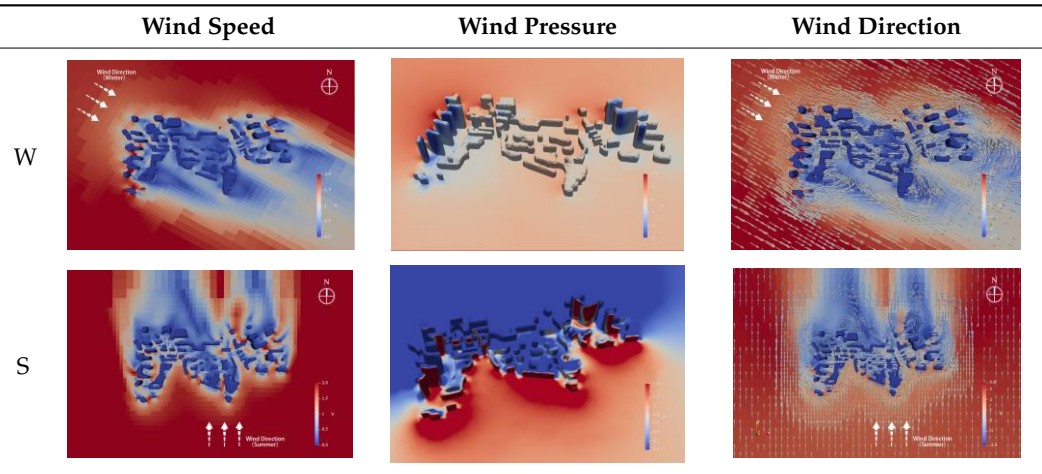

**Table 10.** Rate of change in wind environment evaluation index related to A1A2 settlement layout.

| Seasons\Index | $V_{max}$ | $\overline{V}$ | $R_{max}$ | $\overline{R}$ | $\overline{P}$ |
|---|---|---|---|---|---|
| Winter | 0.069 | 0.012 | 0.069 | 0.012 | 0.012 |
| Summer | −0.050 | −0.044 | −0.050 | −0.044 | −0.083 |

| Seasons\Index | $\sigma$ | $A_{static}$ | $A_{comfort}$ | $A_{strong}$ | $P_{max}$ |
|---|---|---|---|---|---|
| Winter | −0.029 | 0.036 | 0.019 | −0.286 | −0.125 |
| Summer | 0.052 | 0.211 | −0.068 | - | −0.054 |

The overall wind environment quality in winter will decrease, mainly manifested in an increase in the wind velocity value of the site, and the wind velocity will be more susceptible to the influence of buildings. From the perspective of the wind velocity index, significant wind velocity was generated, including the maximum and average wind velocity. The proportion of calm wind areas increases, the proportion of comfort wind increases, and the proportion of strong wind decreases. Meanwhile, the dispersion of wind velocity decreases, and the distribution of wind field tends to be uniform, which is beneficial for improving winter climate conditions. Furthermore, buildings also impact wind velocity, which has a specific adverse impact on the appearance of the building. However, this increase is insignificant and is still within the range the human body can withstand. From the perspective of wind pressure indicators, in winter, the buildings on the site meet the requirements of the specifications, with a decrease in maximum wind pressure and an increase in average wind pressure throughout the site. From the perspective of eddy current generation, compared to the original plan, this plan generates more eddy currents in the area, mainly concentrated in the open square. In addition, the wind environment at the entrance and exit of the site is relatively complex; Ih is also the principal place where vortexes occur.

On the other hand, the quality of the wind environment in summer relatively declined, the proportion of comfortable wind areas decreased, and the number of vortex influences became larger, mainly because of the number of buildings in summer before and after the

building wind pressure difference specification that did not meet the requirements of the specification. From the wind velocity index, the overall mean wind velocity has increased, but the distribution is uneven, and the strong wind velocity is mainly concentrated between the alleyways of multi-story buildings on the windward side. The wind velocity ratio decreases, the influence of buildings on wind velocity is reduced, and the wind pressure of slab buildings is larger. Even so, the proportion of static wind zones increases under this scheme, and the total comfort level decreases; in terms of wind pressure index, it is found statistically that in summer, only eight buildings meet the code requirements, and the nine primary buildings do not meet the code requirements. The maximum wind pressure and the mean wind pressure have decreased. We can see from the vortex cloud chart that the wind distribution is more complicated in the central area between buildings, and at this time, the vortex even shows a straight line in the spacing between buildings.

Table 11 reflects the third option, which is to increase the height of the outer ring building to 33 meters, resulting in a slight change in the wind field. Table 12 reflects the specific numerical indicators under this scenario.

**Table 11.** Option3: A2B1 settlement layout in 1.5 m outgoing wind environment cloud map (W represents winter and S represents summer).

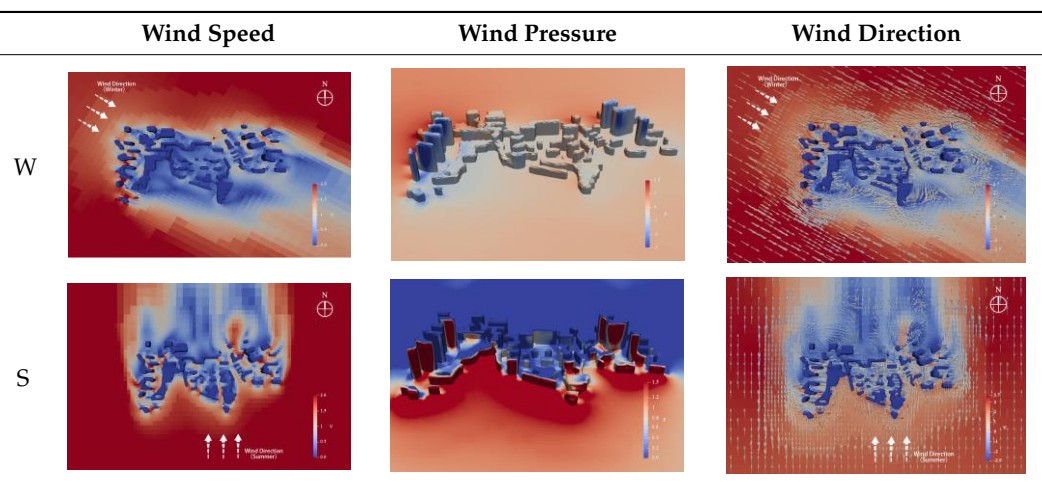

| | Wind Speed | Wind Pressure | Wind Direction |
|---|---|---|---|
| W | | | |
| S | | | |

**Table 12.** Rate of change in wind environment evaluation index related to A2B1 settlement layout.

| Seasons\Index | $V_{max}$ | $\overline{V}$ | $R_{max}$ | $\overline{R}$ | $\overline{P}$ |
|---|---|---|---|---|---|
| Winter | 0.065 | 0.022 | 0.065 | 0.022 | −0.006 |
| Summer | −0.039 | −0.039 | −0.039 | −0.039 | −0.250 |
| Seasons\Index | $\sigma$ | $A_{static}$ | $A_{comfort}$ | $A_{strong}$ | $P_{max}$ |
| Winter | −0.028 | −0.089 | 0.024 | −0.361 | −0.121 |
| Summer | 0.045 | 0.208 | −0.067 | - | −0.054 |

According to simulated statistical data, compared to the original plan, the overall wind environment in winter shows a downward trend. Although the wind field distribution is more uniform, high wind velocity increases, and the degree of influence from buildings increases. The proportion of static wind area decreases, the proportion of strong wind area decreases, and the proportion of comfort zone increases. From the perspective of wind pressure indicators, although the wind pressure difference between the front and rear of the building meets the requirements of the specifications, the average and maximum wind pressure of the building have decreased. The vortex cloud chart shows that compared to the original plan, there is only a tiny amount of vortex influence in winter, but the overall wind environment still needs to be improved.

Compared with the blank control, the overall wind environment in summer has not been significantly improved; the overall wind velocity drop and the uniformity of the wind velocity have decreased, although the degree of building impact on wind velocity has a downward trend. From the wind velocity index, the overall wind velocity situation shows a downward trend and uneven distribution. High wind velocity gradually appears at the corners of various buildings and gradually increases in multi-story building tunnels. However, the proportion of calm wind zones has increased, while the proportion of comfort zones has decreased. From the perspective of wind pressure, there are 11 buildings with a front and rear wind pressure difference higher than 0.5 Pa, which satisfies the "Green Building Evaluation Standard" requirements and has significantly improved compared to the blank control plan. In addition, both the average and maximum wind pressures show a decreasing trend. The vortex cloud map shows that in summer, vortexes are mainly concentrated in point-shaped buildings on open squares, and there is no significant improvement in vortexes compared to open areas.

The layout reflected in Table 13 involves the construction of an 18-meter-high building complex in the outer ring and a 33-meter-high building complex in the inner ring. Table 14 shows the results of the corresponding indicator values.

**Table 13.** Option 4: A1B2 settlement layout in 1.5 m outgoing wind environment cloud map (W represents winter and S represents summer).

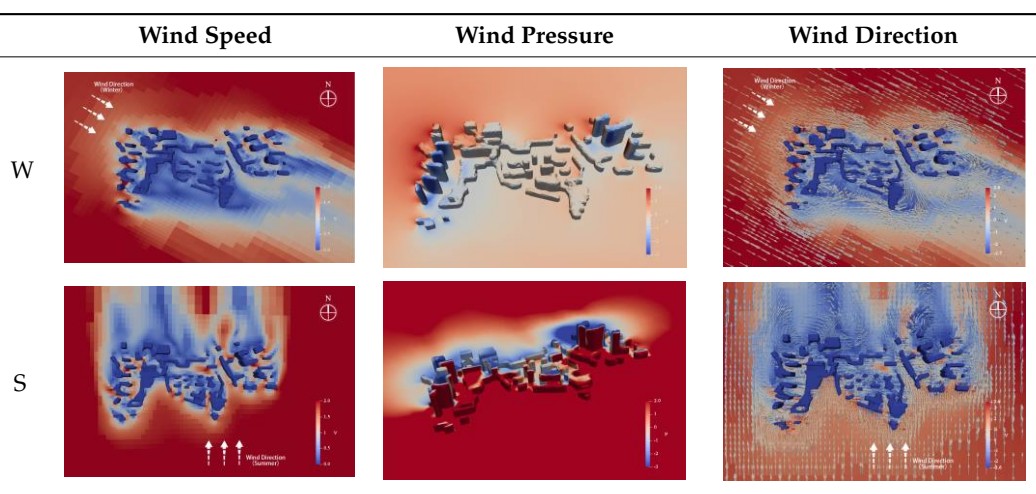

**Table 14.** Rate of change in wind environment evaluation index related to A1B2 settlement layout.

| Seasons\Index | $V_{max}$ | $\overline{V}$ | $R_{max}$ | $\overline{R}$ | $\overline{P}$ |
|---|---|---|---|---|---|
| Winter | 0.062 | 0.022 | 0.062 | 0.022 | −0.081 |
| Summer | −0.052 | −0.036 | −0.052 | −0.036 | −0.345 |

| Seasons\Index | $\sigma$ | $A_{static}$ | $A_{comfort}$ | $A_{strong}$ | $P_{max}$ |
|---|---|---|---|---|---|
| Winter | −0.055 | −0.072 | 0.030 | −0.449 | −0.133 |
| Summer | 0.030 | 0.167 | −0.053 | - | −0.067 |

In winter, the overall wind field environment has improved, and although the overall wind velocity has slightly increased, the prevailing wind field comfort has improved due to uniformity. From the statistical wind velocity index, it is evident that the overall wind velocity has been improved, the maximum wind velocity has been increased, the distribution of the whole wind field has become more uniform, and the proportion of the comfortable wind area has been improved, which indicates that the degree of wind velocity improvement is still in the comfort zone from the wind pressure index, the average wind pressure and maximum wind pressure have decreased, but according to the statistics, the wind pressure difference between the front and rear of the building meets the requirements

in the site. From the vortex cloud diagram, it can be seen that the scheme has a more significant improvement than the original scheme.

In summer, the overall wind velocity within the area is reduced, and the wind velocity distribution within the site is uneven. In terms of the wind velocity index, the maximum wind velocity and the average wind velocity decreased, increasing the proportion of the static wind area, a decrease in the proportion of comfortable wind area, and a decrease in the overall wind velocity distribution uniformity. However, buildings have a more significant impact on wind speed. In terms of the wind pressure index, although the average wind pressure and maximum wind pressure have decreased, the number of buildings with a wind pressure difference between the front and rear of the building meeting the code requirements is 14, which is a significant improvement compared with the original scheme while meeting the code requirements. From the vortex cloud diagram, it can be seen that the influence of the vortex will not increase or decrease under this layout form, and it is still relatively apparent in general.

Table 15 reflects the layout with an inner and outer ring of buildings of 33 m. Table 16 shows the results of the corresponding numerical values of the indicators.

**Table 15.** Option 5: B2B1 settlement layout in 1.5 m out of the wind environment cloud map (W represents winter and S represents summer).

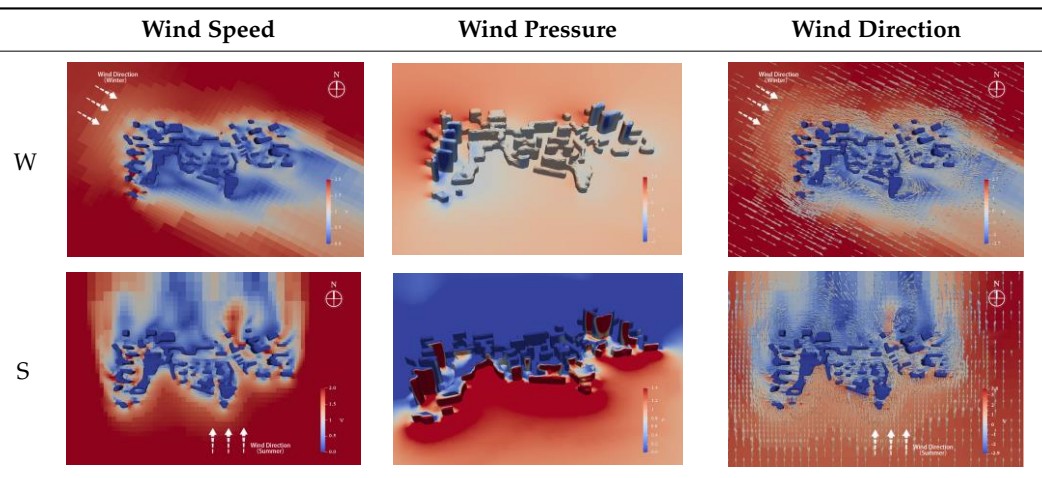

**Table 16.** Rate of change in wind environment evaluation index related to B1B2 settlement layout.

| Seasons\Index | $V_{max}$ | $\overline{V}$ | $R_{max}$ | $\overline{R}$ | $\overline{P}$ |
|---|---|---|---|---|---|
| Winter | 0.063 | 0.021 | 0.063 | 0.021 | 0.004 |
| Summer | −0.038 | −0.041 | −0.038 | −0.041 | −0.317 |
| Seasons\Index | $\sigma$ | $A_{static}$ | $A_{comfort}$ | $A_{strong}$ | $P_{max}$ |
| Winter | −0.034 | −0.071 | 0.026 | −0.389 | −0.121 |
| Summer | 0.048 | 0.194 | −0.062 | - | −0.059 |

When the inner and outer circles of the residence are both 33 m, the wind field in winter is still considerable in general, the proportion of comfort area is significantly increased, and the improvement of wind velocity uniformity is more desirable. The statistical wind velocity index shows that the overall wind velocity increases, but the overall wind velocity distribution within the site is more uniform, the strong wind decreases, and the percentage of static wind area decreases. However, the rate of comfortable wind areas increases. Additionally, the impact of buildings on wind velocity is also more significant. From the wind pressure index, the average wind pressure in winter is slightly higher, while the maximum wind pressure tends to decrease. Except for this, the winter wind farm buildings all meet the wind pressure specification requirements. From the vortex cloud map, it can

be seen that the generation condition of the vortex field in the winter wind field is poor, mainly concentrated in the south entrance direction of the site.

Compared with the original scheme, the summer wind field is significantly lower, the wind velocity is reduced, and the wind velocity uniformity has also decreased. From the wind velocity index, the overall wind velocity has decreased, the uniformity of wind velocity distribution has also deteriorated, the static wind area has increased, and the comfort wind area has decreased, but the wind velocity affected by the buildings has decreased. From the wind pressure index, the difference between the wind pressure before and after for 15 buildings is above 0.5 Pa, which meets the specification requirements and has a better development trend. In addition, the average wind pressure and maximum wind pressure in summer tend to decrease. Compared with the original design, the vortex flow in summer tends to intensify, and it is mainly concentrated in the open space of the point-type residential group and slab-type residential group.

Table 17 reflects the final simulation cloud map of the layout form with 66m height for the outer ring complex and 33m for the inner ring complex. Table 18 reflects the specific values of the corresponding wind environment indicators. The simulation results show that the environmental quality of the wind farm decreases in winter, but the decrease is not large, and in general it is still at a more desirable level.

**Table 17.** Option 6: B2C1 settlement layout in 1.5 m outgoing air environment cloud map (W represents winter and S represents summer).

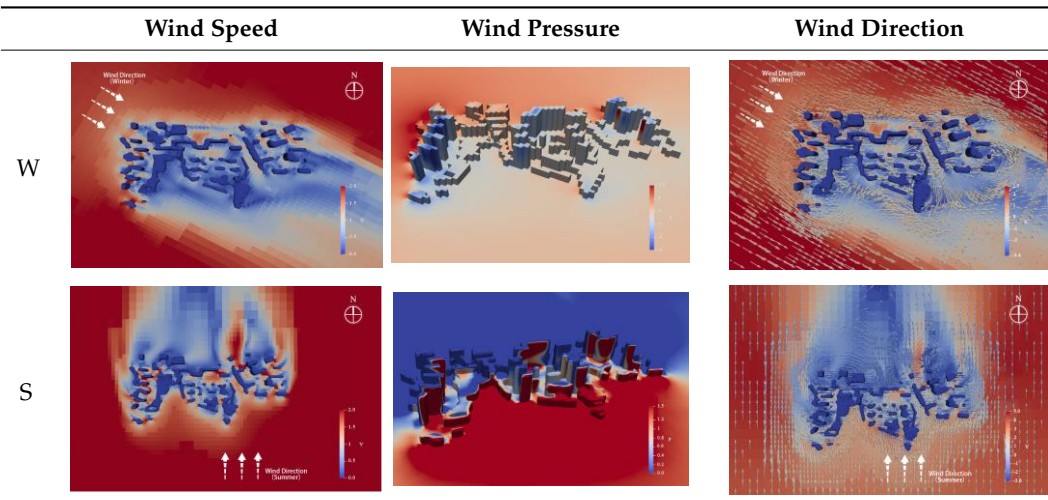

**Table 18.** Rate of change in wind environment evaluation index related to B2C1 settlement layout.

| Seasons\Index | $V_{max}$ | $\overline{V}$ | $R_{max}$ | $\overline{R}$ | $\overline{P}$ |
|---|---|---|---|---|---|
| Winter | −0.004 | 0.000 | −0.004 | 0.000 | 0.050 |
| Summer | 0.022 | −0.048 | 0.022 | −0.049 | 0.120 |
| Seasons\Index | $\sigma$ | $A_{static}$ | $A_{comfort}$ | $A_{strong}$ | $P_{max}$ |
| Winter | −0.004 | 0.047 | −0.009 | 0.132 | 0.014 |
| Summer | 0.067 | 0.236 | −0.076 | - | 0.005 |

Through statistical analysis of wind velocity indices, it is found that the maximum wind velocity has a decreasing trend, and the average wind velocity has not ascended much. The level of impact of buildings on wind velocity has increased, and the wind field is more uniform compared with the original scheme, but there is no significant improvement. The static wind area has increased, but the overall winter comfort has a decreasing trend, which is mainly due to the increase in the proportion of strong wind areas and mainly concentrated in the rear of the slab high rise. From the wind pressure indices, the overall

wind pressure values all show an increasing trend, and the buildings on the site meet the code requirements except in winter. It can be seen from the vortex cloud diagram that the vortex condition of this layout is more desirable compared with the blank control.

The total wind field in summer is slightly reduced compared with the original scheme. From the wind velocity index, in summer, the maximum wind velocity increases, the average wind velocity decreases, the uniformity of wind field distribution also decreases, the wind velocity influenced by buildings decreases, the static wind area accounts for a larger area, and the comfortable wind area accounts for a smaller area. From the wind pressure index, the wind pressure values of the whole environment all show an increasing trend, and there are 14 buildings with a front and rear wind pressure difference above 0.5 Pa, which fully meets the requirements of the specification. For the vortex cloud diagram, the degree of vortex influence also tends to decrease under this layout form.

The height of the inner ring of buildings is 60 meters and the height of the outer ring of buildings is 18 meters. Table 19 reflects the results of the simulated cloud diagrams, while Table 20 reflects the specific numerical results of the wind environment indicators.

**Table 19.** Option 7: A1C2 settlement layout in 1.5 m outgoing wind environment cloud map (W represents winter and S represents summer).

| | Wind Speed | Wind Pressure | Wind Direction |
|---|---|---|---|
| W | | | |
| S | | | |

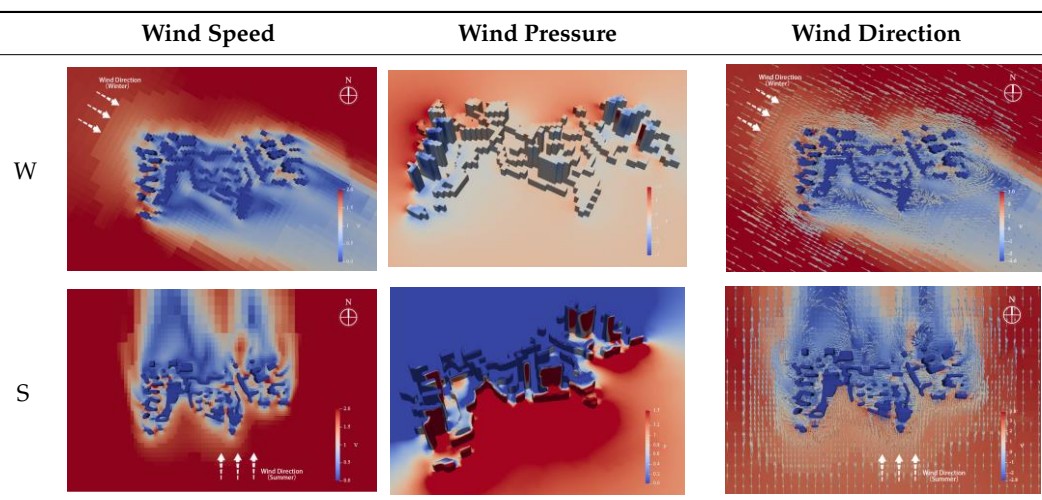

**Table 20.** Rate of change in wind environment evaluation index related to A1C2 settlement layout.

| Seasons\Index | $V_{max}$ | $\overline{V}$ | $R_{max}$ | $\overline{R}$ | $\overline{P}$ |
|---|---|---|---|---|---|
| Winter | −0.018 | −0.019 | −0.018 | −0.019 | 0.047 |
| Summer | −0.035 | −0.060 | −0.035 | −0.060 | −0.132 |

| Seasons\Index | $\sigma$ | $A_{static}$ | $A_{comfort}$ | $A_{strong}$ | $P_{max}$ |
|---|---|---|---|---|---|
| Winter | 0.017 | 0.119 | −0.015 | 0.230 | −0.001 |
| Summer | 0.100 | 0.277 | −0.089 | - | −0.016 |

During the winter, the overall wind field was improved to some extent, but strong winds still existed at the site, impacting the wind environment's comfort. From the values of the wind velocity index, the overall wind velocity in the site showed a decreasing trend, the proportion of static wind area increased, and the proportion of comfort increased, but the overall uniformity also showed a decreasing trend. In addition, the wind velocity affected by the building decreased. On the wind pressure index, the average wind pressure in winter slightly increased and the highest wind pressure slightly decreased. In addition, the wind pressure difference between the front and rear of the building in the site in winter meets the specification. It can be seen from the vortex cloud diagram that the improvement effect of a vortex is evident in winter.

During the summer months, there is a tendency for the overall wind field to weaken. From the wind velocity index, the wind velocity in each region shows a downward trend, increasing the proportion of calm wind areas and decreasing the proportion of comfortable wind areas. Even worse, on this site, the uniformity of wind velocity has dramatically decreased. However, due to the influence of buildings, the wind velocity has also decreased significantly. From the perspective of wind pressure indicators, 13 buildings have achieved wind pressures above 0.5 Pa before and after the building, which has improved to varying degrees compared to the original plan, but the overall wind pressure has a downward trend. From the vortex flow cloud diagram, this arrangement method also has a significant improvement effect on the vortex flow cloud diagram.

When the height of the outer ring of the building is 33 meters and the height of the inner ring of the building is 60 meters, the final results based on the responses of Tables 21 and 22 show that there is a significant tendency for the winter wind field to increase under this layout condition.

**Table 21.** Option 8: B1C2 settlement layout in 1.5 m outgoing wind environment cloud map (W represents winter and S represents summer).

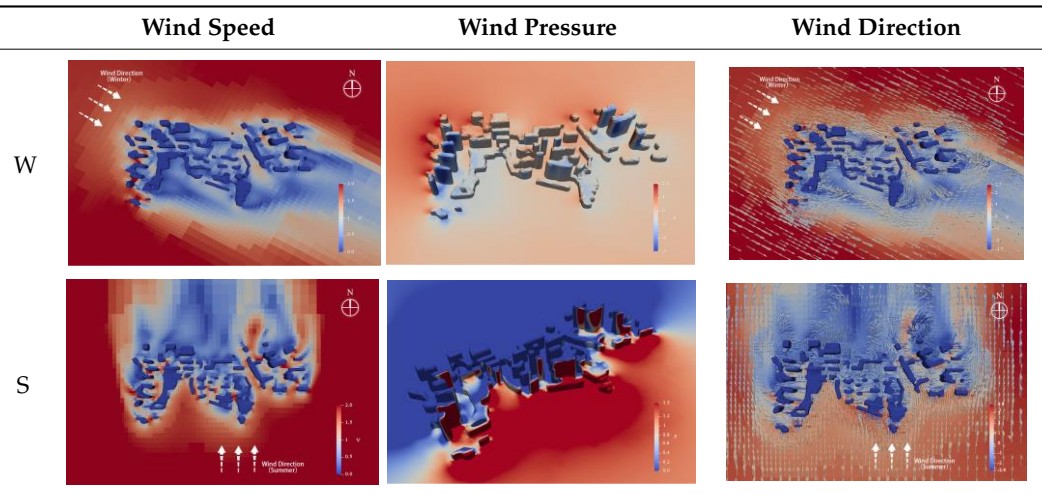

| | Wind Speed | Wind Pressure | Wind Direction |
|---|---|---|---|
| W | | | |
| S | | | |

**Table 22.** Rate of change in wind environment evaluation index related to B1C2 settlement layout.

| Seasons\Index | $V_{max}$ | $\overline{V}$ | $R_{max}$ | $\overline{R}$ | $\overline{P}$ |
|---|---|---|---|---|---|
| Winter | 0.064 | 0.032 | 0.064 | 0.033 | −0.012 |
| Summer | −0.057 | −0.005 | −0.057 | −0.004 | −0.211 |

| Seasons\Index | $\sigma$ | $A_{static}$ | $A_{comfort}$ | $A_{strong}$ | $P_{max}$ |
|---|---|---|---|---|---|
| Winter | −0.047 | −0.134 | 0.036 | −0.476 | −0.126 |
| Summer | 0.008 | 0.009 | −0.004 | - | −0.025 |

From the wind velocity index, the overall wind velocity tends to increase in winter, but the wind velocity uniformity increases, the static wind area decreases, and the proportion of comfortable wind area increases. Additionally, the impact of buildings on wind velocity also tends to increase. The wind pressure index shows that the mean wind pressure in the wind field has an increasing trend, while the maximum wind pressure has a decreasing trend. Other than that, the wind pressure difference between the front and rear of the building in the winter site meets the code requirements. From the vortex cloud diagram, it can be seen that this arrangement will have a more significant vortex field distribution in winter.

Compared with the original plan, the overall wind environment in summer has decreased. From the perspective of wind velocity indicators, the overall wind velocity is

trending downwards, with an increase in the proportion of calm wind area and a decrease in the proportion of comfort. However, compared to the original plan, the on-site wind velocity distribution is more uneven. Furthermore, the impact of buildings on wind velocity has also decreased. According to statistical wind pressure indicators, 13 buildings meet the requirement of a pressure difference greater than 0.5 Pa before and after construction in summer. Although the overall wind pressure of the site shows a decreasing trend, there is a specific improvement compared to the blank control. The vortex cloud diagram shows that the wind field environment is more complex in summer, with apparent vortex fields in front of open panel buildings and behind point buildings.

In the last case, the height of all buildings on the base is 60 meters. The results are shown in Tables 23 and 24.

**Table 23.** Option 9: C1C2 settlement layout in 1.5 m outgoing wind environment cloud map (W represents winter and S represents summer).

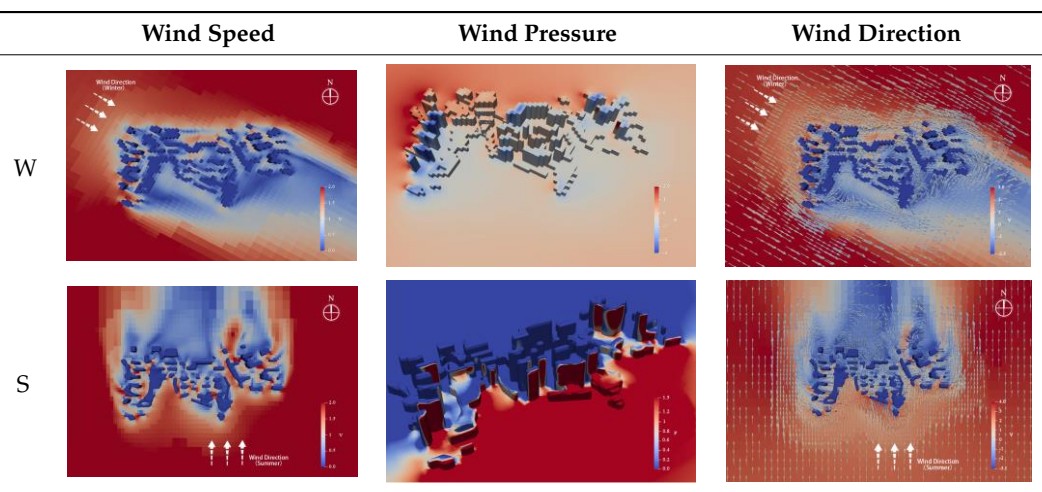

**Table 24.** Rate of change in wind environment evaluation index related to C1C2 settlement layout.

| Seasons\Index | $V_{max}$ | $\overline{V}$ | $R_{max}$ | $\overline{R}$ | $\overline{P}$ |
|---|---|---|---|---|---|
| Winter | −0.003 | 0.005 | −0.003 | 0.006 | 0.028 |
| Summer | −0.023 | −0.039 | −0.023 | −0.038 | 0.050 |

| Seasons\Index | $\sigma$ | $A_{static}$ | $A_{comfort}$ | $A_{strong}$ | $P_{max}$ |
|---|---|---|---|---|---|
| Winter | −0.015 | 0.015 | −0.009 | 0.132 | 0.007 |
| Summer | 0.060 | 0.170 | −0.054 | - | 0.029 |

The winter wind environment generally decreases in this layout form, but the decrease is insignificant, so it is still in an ideal state. The numerical wind velocity index shows that the wind velocity of the winter wind field has increased compared with the blank control, but the maximum wind velocity has decreased. Overall, the uniformity of the wind field has been improved. The static wind area has increased, but the proportion of comfortable wind has decreased. In addition, the buildings' influence on the whole site's wind velocity increases, but the influence on the maximum wind velocity decreases. From the wind pressure index, all the wind pressure values in the wind field increase, and the maximum wind pressure values also increase. Other than that, the buildings on the winter site meet the wind pressure code requirements. From the vortex cloud diagram, it can be seen that this arrangement creates a high-impact vortex near the entrance in wintertime.

Compared with the original scheme, the summer wind here is significantly weaker. From the wind velocity index, the wind velocity of the whole site shows a decreasing trend influenced by the buildings, while the trend of wind velocity uniformity decreases more obviously inside the site. The proportion of the static wind zone increases sharply, while

the proportion of the comfort zone decreases. According to the wind pressure index, there are 12 buildings that meet the specification of a wind pressure difference between the front and rear of the building is greater than 0.5 Pa in summer, which is some improvement compared with the original scheme. The vortex cloud map shows that the wind field in the whole area is complicated in summer, and there are apparent vortexes behind the butterfly-type buildings.

## 5. TOPSIS-Entropy Weight Method Comprehensive Evaluation Model

### 5.1. Model Establishment

According to the above discussion, there are ten indicators divided into winter and summer conditions, so there are a total of 20 indicator items from which appropriate indicators are selected, and the nature of each indicator is judged to establish the evaluation model of TOPSIS.

The first step is to identify the evaluation indicators needed. First, we need to select the wind velocity index, the determined evaluation index that includes the maximum wind velocity and the proportion of comfortable wind area. Secondly, it is necessary to evaluate the index of wind field uniformity, namely wind velocity dispersion. Then, the maximum wind velocity ratio, an evaluation index for evaluating wind velocity affected by buildings, is also included. As for the evaluation index of wind pressure, we select the number of buildings with the difference in wind pressure before and after winter and summer buildings that meet the standards as an evaluation index. The values of each indicator for all programs were counted and the results are shown in Table 25.

**Table 25.** Numerical results of statistical simulation calculations.

| Option\ Index | Winter | | | | | Summer | | | | |
| --- | --- | --- | --- | --- | --- | --- | --- | --- | --- | --- |
| | $R_{max}$ min | $\sigma$ min | $A_{comfort}$ max | $A_{static}$ min | $V_{max}$ min | $R_{max}$ min | $\sigma$ min | $A_{comfort}$ max | $V_{max}$ max | the number of $p > 0.5$ Pa max |
| Option 1 | 1.135 | 0.616 | 0.956 | 0.357 | 2.326 | 1.068 | 0.733 | 0.706 | 2.521 | 8 |
| Option 2 | 1.130 | 0.616 | 0.960 | 0.314 | 2.316 | 1.080 | 0.728 | 0.707 | 2.550 | 11 |
| Option 3 | 1.061 | 0.634 | 0.938 | 0.345 | 2.175 | 1.124 | 0.696 | 0.757 | 2.654 | 10 |
| Option 4 | 1.127 | 0.599 | 0.966 | 0.320 | 2.310 | 1.066 | 0.717 | 0.717 | 2.517 | 14 |
| Option 5 | 1.127 | 0.613 | 0.962 | 0.320 | 2.311 | 1.082 | 0.730 | 0.710 | 2.553 | 15 |
| Option 6 | 1.057 | 0.632 | 0.930 | 0.361 | 2.167 | 1.149 | 0.743 | 0.700 | 2.711 | 14 |
| Option 7 | 1.042 | 0.645 | 0.924 | 0.386 | 2.136 | 1.085 | 0.766 | 0.690 | 2.560 | 13 |
| Option 8 | 1.125 | 0.602 | 0.963 | 0.313 | 2.306 | 1.083 | 0.749 | 0.697 | 2.557 | 13 |
| Option 9 | 1.058 | 0.625 | 0.930 | 0.350 | 2.169 | 1.099 | 0.738 | 0.716 | 2.593 | 12 |

Step 2: determine the type of each indicator.

(1) Maximum wind velocity: in winter, it is a minimum indicator, which is better when smaller; in summer, the maximum indicator is better when larger, and contributes to the formation of natural ventilation.

(2) Comfortable wind velocity zone, which is the maximum indicator. The larger the value, the better the solution.

(3) Wind velocity dispersion: this indicator belongs to the minimum statistical type in both winter and summer; the larger the value, the more uniform the wind field is, and the higher the corresponding comfort level is.

(4) Maximum wind velocity ratio: this indicator is a minimum statistical indicator; that is, within the comfort zone, the less the wind velocity is affected by the building, the better.

(5) Wind pressure: The indicator represents the number of buildings within the site that meet the specifications; the more extensive the value, the higher the evaluation. Since the wind pressure difference between the front and rear of all the buildings in the

winter site is within a reasonable range, the indicator can be omitted when the input evaluation index is carried out.

Step 3: The entropy weighting method determines the weights of each indicator as follows:

The entropy weight method provides a weight basis for TOPSIS, and the results obtained are shown in Table 26. The maximum wind speed in winter has the smallest entropy value, and its information utility value is the highest, so its weighting coefficient is the highest.

**Table 26.** Entropy weighting method to determine the value of index weights.

| | **Winter** | | | | | **Summer** | | | | |
|---|---|---|---|---|---|---|---|---|---|---|
| | $R_{max}$ | $\sigma$ | $A_{comfort}$ | $A_{static}$ | $V_{max}$ | $R_{max}$ | $\sigma$ | $A_{comfort}$ | $V_{max}$ | *the number of p > 0.5 Pa* |
| Weights | 0.284 | 0.1173 | 0.0001 | 0.1037 | 0.2877 | 0.0882 | 0.1102 | 0.0002 | 0.0001 | 0.0085 |

*5.2. Model Solving*

The data were processed using MATLAB 2020 software, and the following results were obtained: Table 27 represents the matrix of forwarding and Table 28 reacts to the scores of each scenario; we ranked the scores of each scenario and the final results are shown in Figure 6. The code for this operation can be found in Figure S1 in the Supplementary Materials.

**Table 27.** Matrix after normalization.

| | **Winter** | | | | | **Summer** | | | | |
|---|---|---|---|---|---|---|---|---|---|---|
| Option\Index | $R_{max}$ | $\sigma$ | $A_{comfort}$ | $A_{static}$ | $V_{max}$ | $R_{max}$ | $\sigma$ | $A_{comfort}$ | $V_{max}$ | *the number of p > 0.5 Pa* |
| Option 1 | 0.4555 | 0.1282 | 0.3299 | 0.2665 | 0.4557 | 0.135 | 0.6204 | 0.3547 | 0.3429 | 0.2688 |
| Option 2 | 0.0000 | 0.338 | 0.3362 | 0.1885 | 0.0000 | 0.4375 | 0.2925 | 0.3308 | 0.3257 | 0.2150 |
| Option 3 | 0.0308 | 0.338 | 0.3376 | 0.468 | 0.0302 | 0.3727 | 0.3368 | 0.3313 | 0.3294 | 0.2957 |
| Option 4 | 0.0492 | 0.5362 | 0.3397 | 0.429 | 0.0483 | 0.4483 | 0.4343 | 0.336 | 0.3252 | 0.3763 |
| Option 5 | 0.0492 | 0.373 | 0.3383 | 0.429 | 0.0453 | 0.3619 | 0.319 | 0.3327 | 0.3298 | 0.4032 |
| Option 6 | 0.4801 | 0.1515 | 0.3271 | 0.1625 | 0.4798 | 0.0000 | 0.2038 | 0.328 | 0.3502 | 0.3763 |
| Option 7 | 0.5725 | 0.0000 | 0.325 | 0.0000 | 0.5734 | 0.3457 | 0.0000 | 0.3233 | 0.3307 | 0.3494 |
| Option 8 | 0.0616 | 0.5012 | 0.3387 | 0.4745 | 0.0604 | 0.3565 | 0.1507 | 0.3266 | 0.3303 | 0.3494 |
| Option 9 | 0.474 | 0.2331 | 0.3271 | 0.234 | 0.4738 | 0.2701 | 0.2481 | 0.3355 | 0.3350 | 0.3226 |

The scores are summarized as follows:

**Table 28.** Final scores of the TOPSIS-entropy weighting method for each scenario.

| | **Option 1** | **Option 2** | **Option 3** | **Option 4** | **Option 5** | **Option 6** | **Option 7** | **Option 8** | **Option 9** |
|---|---|---|---|---|---|---|---|---|---|
| Score | 0.1539 | 0.0711 | 0.084 | 0.0987 | 0.0848 | 0.1346 | 0.1336 | 0.0881 | 0.1511 |

The final scores were visualized with the following:

## Score of each scheme after TOPSIS evaluation

| | Option 2 | Option 3 | Option 5 | Option 8 | Option 4 | Option 7 | Option 6 | Option 9 | Option 1 |
|---|---|---|---|---|---|---|---|---|---|
| ■ Scores | 0.0711 | 0.084 | 0.0848 | 0.0881 | 0.0987 | 0.1336 | 0.1346 | 0.1511 | 0.1539 |

**Figure 6.** Ranking and scores of the programs obtained according to the TOPSIS comprehensive evaluation method.

The TOPSIS method ranks the evaluation objects' distances from the positive and negative ideal solutions to evaluate their advantages and disadvantages. Based on the above discussion, the order of merit of each option is option 1 > option 9 > option 6 > option 7 > option 4 > option 8 > option 5 > option 3 > option 2.

### 5.3. K-Means Clustering and Systematic Clustering

Design decision making is a complex compromise process. Although there are only nine options to consider and compare the advantages and disadvantages of each option comprehensively and to conduct a comparative analysis, in this research, the clustering algorithm is used to mark the options of the final TOPSIS score, from which representative solutions are extracted and from which the standard and similar points among them are identified. The clustering results under different k values were compared using the K-means clustering method in SPSS software. The results showed that when the k value was two, there were significant differences in block morphology between different clustering groups, and the characteristics of block morphology within the same clustering group were the same. Based on this, the two were verified using the method of systematic clustering, and the results showed that both of them could explain each other. On this basis, the k value of two was defined. Table 29 and Figure 7 present the final results obtained from the two clustering methods.

**Table 29.** K-means clustering results and distances.

| Programs | Clustering | Distance |
|---|---|---|
| 1 | 1 | 0.011 |
| 2 | 2 | 0.014 |
| 3 | 2 | 0.001 |
| 4 | 2 | 0.013 |
| 5 | 2 | 0.001 |
| 6 | 1 | 0.009 |
| 7 | 1 | 0.010 |
| 8 | 2 | 0.003 |
| 9 | 1 | 0.008 |

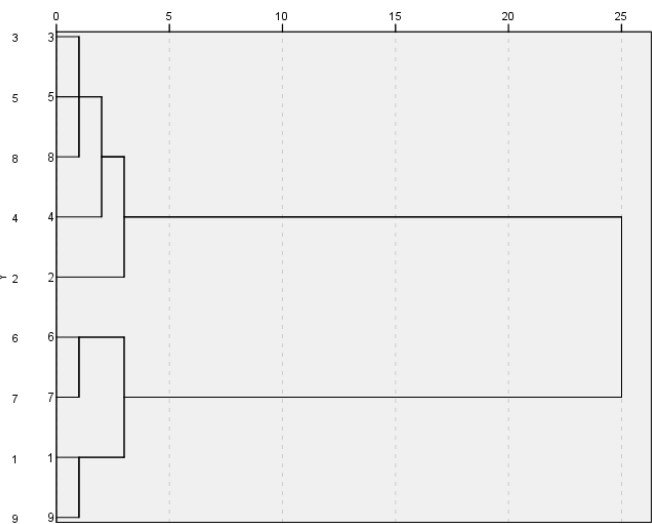

**Figure 7.** Spectral map obtained by systematic clustering.

*5.4. Conclusions and Discussion*

According to the results of clustering, as shown in Table 30, it is easy to see that in this layout form, to create a more comfortable wind environment, high-rise buildings should be designed with more consideration, especially the northeast, northwest, and north-oriented building groups of the building complex in a staggered layout of high-rise buildings can effectively block the cold winds in winter, when the internal arrangement of low-rise, mid-rise, and high-rise buildings can relatively form a more considerable wind environment. In addition, the layout of medium and low-rise buildings in the area's northeast, northwest, and north directions may lead to a relatively poor wind environment. Still, the internal layout of point-like high-rise buildings can effectively mitigate it.

**Table 30.** Clustering results.

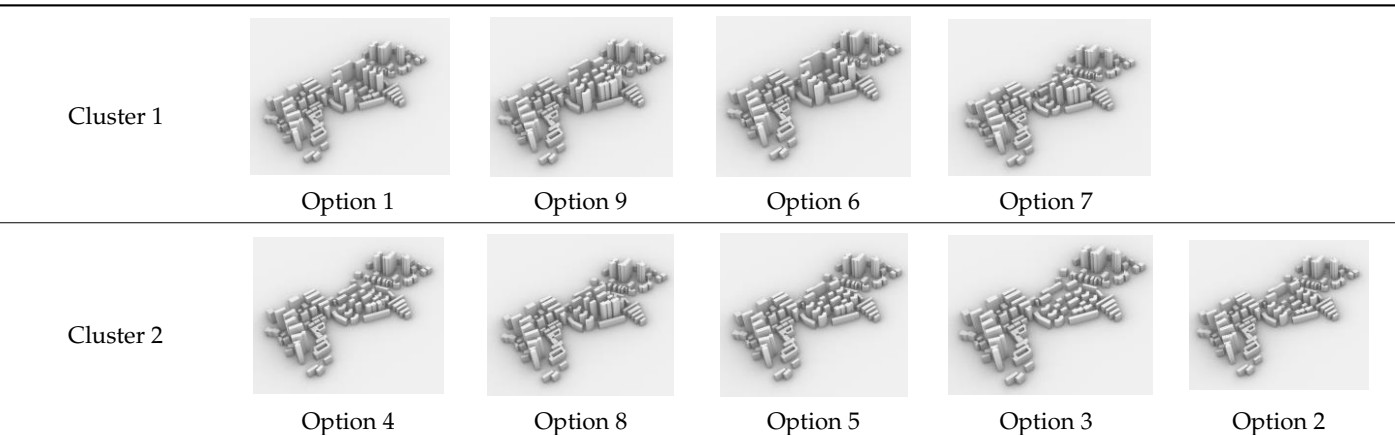

| | | | | | |
|---|---|---|---|---|---|
| Cluster 1 | Option 1 | Option 9 | Option 6 | Option 7 | |
| Cluster 2 | Option 4 | Option 8 | Option 5 | Option 3 | Option 2 |

## 6. Conclusions and Discussion

*6.1. Discussion*

(1)  Three-dimensional modeling analysis of the influence of surrounding buildings on the site buildings

When performing 3D modeling and numerical simulation, the surrounding buildings will impact the buildings on the site. The magnitude of the changes of each indicator is summarized in the below figure.

According to the statistical data, as shown in Figure 8, we can see that the deviation produced for winter is more significant than that for summer when the numerical simulation is performed without considering the influence of the surrounding buildings: the

combined cumulative effect of the wind field environment generated in winter on wind indicators is 3.24, while the combined cumulative variation of each indicator in summer is only 1.71.

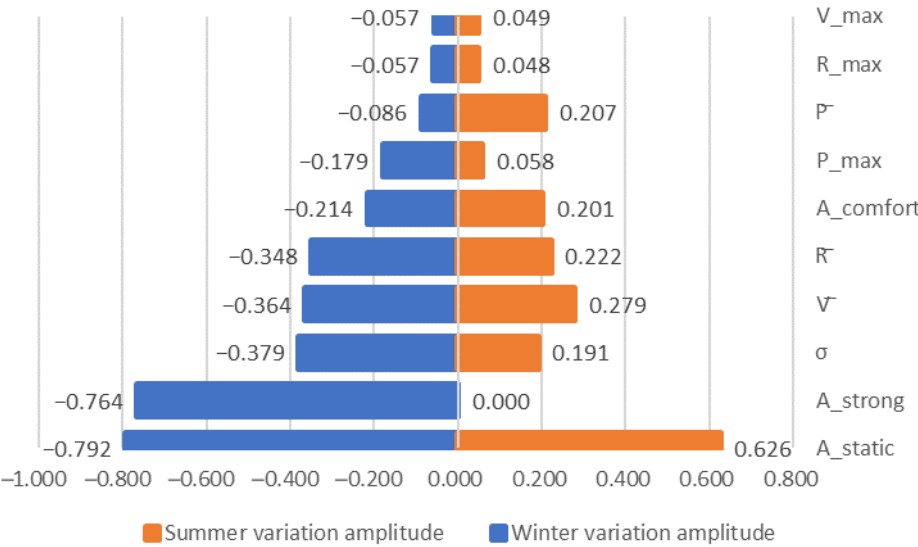

**Figure 8.** Visualization of the magnitude of change in each wind evaluation index in winter and summer.

On the other hand, if the influence of surrounding buildings is not considered, the simulation results in summer will be overestimated. The general tendency is that the building strengthens the wind velocity, but the wind velocity influenced by the building is decreasing, the mean wind pressure and wind velocity are biased towards a relatively more comfortable range, and the uniformity of the wind field is improved.

(2)    Shortcomings and prospects:

The values obtained through software simulation in this study still need to be compared with the results measured at the site to verify the accuracy of the results. The design objectives of this study do not take into account factors such as the rotation angle of settlements, building density and floor area ratio, and subsequent studies can carry out multi-objective optimization solving based on this. The research is not sufficiently accurate, and plug-ins such as Ladybug, due to the limited time of research and development, have a simulation accuracy and computing efficiency that is different from the traditional wind-heat environment simulation software, and subsequent studies can consider factors such as urban subsurface materials, vegetation, and water bodies. In addition, the simulation of building performance optimization consumes a huge amount of time and cost, and subsequent research can take advantage of today's emerging machine learning technology to achieve rapid prediction of building performance simulation, which in turn can assist designers and researchers in quantifying the analysis scheme and optimizing the design work more efficiently.

### 6.2. Conclusions

This paper takes the Xishan Huijing neighborhood in Changsha, Hunan Province, as an example and numerically simulates the building layout of the area with different height distributions based on Butterfly 0.0.05 software, statistically analyzes the wind velocity, wind pressure, vortex, wind velocity ratio, wind velocity dispersion, wind area ratio situation, and other indicators, and visualizes them. The optimal solution is explored using the entropy-based TOPSIS method and K-means clustering. Finally, it is concluded that the best layout form of the district is 60 m in height for the northeast, northwest, and north-oriented building clusters and 18 m in height for the inner circle building clusters, which

can satisfy the requirements of wind environment comfort in the context of environmental liability to the greatest extent. Based on the above discussion and simulation study, the following conclusions are drawn:

(1) Concerning the degree of influence of the surrounding buildings, the deviation of the integrated various indicators produced in winter when analyzing the existing residential complex in the area without considering the surrounding buildings would be greater than that in summer, with the overall magnitude of deviation close to more than two times. In addition to that, not considering the influence of the surrounding buildings on the site would make the simulation results in summer be overestimated.

(2) In terms of the weight of each wind environment index, the results of the index weights determined by the entropy-based TOPSIS method indicate that wind protection in winter should be given priority over ventilation consideration in summer for wind environment design in the region; among them, the maximum wind velocity in winter has the most significant weight, reaching 0.287. On the other hand, in the summer ventilation design, the uniformity of the wind field is the most crucial index that should be considered. Therefore, when studying the wind environment of the region in the future, suitable evaluation indices for winter and summer can be selected according to this conclusion.

(3) As for the matching of building heights, in this layout form, the northeast, northwest, and north-oriented arrangement of high-rise buildings can bring a more comfortable wind environment to the site, which is the conclusion obtained by combining various wind environment evaluation indices. The form of a multi-story and low-rise building combination is not particularly suitable for the formation of a comfortable wind field environment under this area.

**Supplementary Materials:** The following supporting information can be downloaded at: https://www.mdpi.com/article/10.3390/su151612480/s1, Figure S1: TOPSIS entropy weight method related codes; Table S1: Butterfly Table Settings.

**Author Contributions:** Methodology, X.L., W.W. and J.S.; Numerical simulation, X.L. and J.S.; Manuscript writing, X.L. and Z.W.; Picture editing, X.L. and K.L.; and Conceptualization, X.L. All authors have read and agreed to the published version of the manuscript.

**Funding:** This research received no external funding.

**Institutional Review Board Statement:** Not applicable.

**Informed Consent Statement:** Not applicable.

**Data Availability Statement:** Not applicable.

**Acknowledgments:** The data presented in this study are available upon request from the first author.

**Conflicts of Interest:** The authors declare no conflict of interest.

## Nomenclature

| | |
|---|---|
| $V_{max}$ | Maximum wind velocity |
| $\overline{V}$ | Average wind velocity |
| $R_{max}$ | Maximum wind velocity ratio |
| $\overline{R}$ | Average wind velocity ratio |
| $\overline{P}$ | Average wind pressure |
| $P_{max}$ | Maximum wind pressure |
| $A_{strong}$ | Percentage of strong wind area |
| $A_{comfort}$ | Percentage of comfort air area |
| $A_{static}$ | Percentage of static wind area |
| $\sigma$ | Wind velocity dispersion |

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
