# Peer review of "Simulation Study on Outdoor Wind Environment of Residential Complexes in Hot-Summer and Cold-Winter Climate Zones Based on Entropy-Based TOPSIS Method"

_sustainability, doi:10.3390/su151612480_

Round 1
Reviewer 1 Report
Using a case study, this paper intends to identify the most suitable outdoor wind environment for residential areas in China's hot summer and cold winter regions. The study results intend to be the basis for the optimized design of residential buildings in this type of climate, strengthening natural ventilation. The study's results have the potential to be helpful for future building settlements' planning, construction and spatial arrangement.
It is an interesting study within a well-known topic, but its innovative character is satisfactorily presented. In this context, the list of the innovative aspects of the paper should be moved from the Conclusions to the Introduction. Instead, in the Conclusion, the innovations should be clearly evidenced. It would also be relevant to discuss deeper the reliability of the results and the extrapolation of the conclusions to other situations together with the limitations of the proposed method. The application of the entropy weighting method must be better discussed and not presented as a black box. More details are necessary as, for example, on the weighting factors used, among others. Figures must be improved as some are not readable.
Minor English editing is needed.
Reviewer 2 Report
The paper is well argued.
The research is of great interest and contemporaneity, and the TOPSIS method is very useful to relate from the use of parametric digital tools to the discussion of indicators. The relevant elements of innovation reported in the "conclusion paragraph" offer a discussion to the advancement of research at the international level on the topic of Simulation Study on Outdoor Wind Environment .
In the research question addressing the capabilities of using parametric digital tools for environmental analysis, the researchers' contribution appears to be of interest, due to the choice of considering a scenario application in extreme climate, through the effect of wind flows in hot summer and cold winter. Compared with other material applied in the field, the choice of "cluster" methodology and the application of Topsis used n entropy regime, more carefully defines the experimentation on residential buildings and the use of such indicator system for the pre-design phase. The research diagram for the framework, connects well the use of the tools with the data input and output process and the climate analysis performed.
In the simulation activities of the "data analysis" , options could include considering a few cases with elements of "obstruction" to the incidence of winds, because this is what can happen at the residential cluster level and would have influence for the "funnel effect" and the "canyon effect" and then expanding the Topsis methodology also to this component of calculation and simulation and adding the results in the tables, figures and tracing new references on the applications already conducted internationally. The conclusion paragraph answers the research question well in terms of possible innovation in the relationship between indicator structure, use of tools and testing methodologies for pre-design on residential buildings.
Reviewer 3 Report
In this paper a simulation study on outdoor wind environment of residential complexes in hot summer and cold winter climate zones based on TOPSIS Entropy Weight Method - Taking Changsha Xishan Huijing Community, as an example, is made. Before being published, I suggest some improvements in the presentation and in the content.
The figures should be improved. Increase the dimensions and increase the quality, as example. In figure 2, as example, the text is not understandable.
More details about the numerical CFD model should be added.
All inputs of each numerical simulation should be added, as example, in a table.
The grid generation, used in the numerical simulation, should be added in a 2D or 3D picture.
Add, in each numerical simulation, a picture with the wind direction. Add also the northeast, northwest, and northoriented building direction.
More information about the meshing and grid independency should be added.
More details about the future works should be also added.
The future research should be more improved.
Moderate editing of English language required
Reviewer 4 Report
the abstract gives a broad overview of the study's objectives and methodology but lacks crucial details, such as the specific methods used in each step and quantitative results. The novelty of the research and its contributions are not clearly highlighted, and the conclusions are somewhat vague and lack specific evidence from the study's findings. Providing more details, quantitative data, and explicit statements about the novelty and implications of the research would improve the abstract significantly.
1. Lack of Clear Objectives: The introduction mentions the increase in urbanization and its impact on wind environments. However, it does not explicitly state the specific objectives of the research. The introduction should clearly outline the purpose and goals of the study.
2. Repetition and Wordiness: Some parts of the introduction seem repetitive and wordy. It would be beneficial to streamline the text and eliminate unnecessary repetition to improve clarity and conciseness.
3. Citation References: As previously mentioned, the introduction references several works (e.g., [1], [2], [3]) but does not include a proper reference list. Including a complete reference list is essential for academic integrity and proper acknowledgment of prior research.
4. Missing Gap in Knowledge: The introduction could be strengthened by highlighting the existing gap in knowledge or identifying the limitations of previous research in addressing the wind environment optimization for residential areas in hot-summer and cold-winter regions.
5. Lack of Novelty Statement: As mentioned earlier, the introduction does not explicitly state the novelty or unique contribution of the research. It should provide a clear statement on what new insights or advancements this study brings to the field.
6. Specific Research Context: The introduction mentions various studies related to wind environment optimization, but it would be helpful to provide more context and relevance to the current research.
7. Clear Transition to Methodology: The introduction should ideally provide a smooth transition to the methodology section, explaining how the research aims to address the identified gaps and objectives.
To improve the introduction, consider addressing the points mentioned above, particularly providing a clear statement of objectives, identifying the research gap, and explaining the novelty of the study. Additionally, make sure to provide a proper reference list for the cited works and ensure that the introduction flows logically into the methodology section.
The "Materials and Methods" section
1. Inconsistent Subsections: The subsections in the "Materials and Methods" section are not consistently numbered. For example, while the subsection "Selection of wind environment evaluation indexes" is numbered correctly, the subsections "Overview of TOPSIS-entropy method" and "Introduction of simulation tools" are not numbered.
2. Lack of Explanation for Index Selection: The section briefly mentions the selection of five wind environment evaluation indexes, but it lacks a rationale or explanation for choosing these specific indexes. Providing a brief justification for each index's relevance to the research would be beneficial.
3. Ambiguity in Equations: In the "Wind velocity dispersion" index description, the equation is provided, but it lacks clear labeling of variables (e.g., x_i, n, etc.). Adding clear labels and explanations for the variables in the equation would improve understanding.
4. No Definition for Wind Comfort Zones: The terms "Silent wind," "Comfortable wind," and "Severe winds" are mentioned under the "Wind area ratio" index, but they are not defined. Readers may not be familiar with these terms, and providing clear definitions would be helpful.
5. Clarification on Combined Evaluation Method: The section mentions combining the entropy-based TOPSIS method to evaluate wind environment indexes in different seasons and building heights. However, it does not explain how these two methods will be integrated or the specific steps involved in the combined evaluation.
6. Specific Parameters for Simulation: The section introduces the simulation tools used (Rhino, Butterfly, Open Foam, Paraview) but does not provide specific parameters or settings used for the simulations. Including essential simulation parameters would enhance the transparency and reproducibility of the research.
7. Clarity on Workflow: While the main workflow is mentioned, it lacks specific details on how each software tool is utilized and how the data flows between them. Providing a step-by-step breakdown of the workflow would make it easier for readers to follow the research process.
8. Clarification on the Ladybug Plug-in: The section mentions the Ladybug plug-in developed by Mostapha at the University of Pennsylvania, but it does not explain its role in the simulation process or how it connects with Open Foam and Rhino.
To improve the "Materials and Methods" section, consider addressing the points mentioned above. Providing clear explanations for the selection of wind evaluation indexes, defining terms, and specifying simulation parameters would enhance the section's clarity. Additionally, detailing the steps involved in the combined evaluation method and providing a more thorough explanation of the workflow with software tools would improve the reproducibility of the research.
Minor editing of English language required
Reviewer 5 Report
The paper has good piece of research and some interesting findings. The authors prepared a well-organized article. However, it is a very long article. As it stands, it seems difficult to read and follow. It should be shorter and more concise. I have some minor revisions.
I have some comments and suggest as following:
- The title is too long. A shorter, clearer and more inclusive title should be given.
- More concrete results of the study should be given in the abstract section.
- The importance of the study, its difference from the literature and its original aspect should be stated at the end of the introduction.
- In the results section where the findings are given, the results obtained should be given by discussing with the literature.
- In the conclusion part, the effects of the study on researches in other fields should be explained.
Round 2
Reviewer 1 Report
The authors satisfactorily addressed the reviewers' comments.
Minor editing of English language required
Reviewer 3 Report
In the actual version, in general, all suggestions given by the reviewer was commented.
Minor editing of English language required
Reviewer 4 Report
Thank you for revised version, would be pleased to do just a few language editing.
BEST
Minor editing of English language required